# Unraveling Rising Mortality: Statistical Insights from Japan and International Comparisons

**DOI:** 10.3390/healthcare13111305

**Published:** 2025-05-30

**Authors:** Hiroshi Kusunoki

**Affiliations:** Department of Internal Medicine, Osaka Dental University, 8-1 Kuzuhahanazonocho, Hirakata 573-1121, Osaka, Japan; kusunoki1019@yahoo.co.jp

**Keywords:** healthcare system, declining population, demographic shifts, COVID-19 pandemic, mortality

## Abstract

Since the onset of the COVID-19 pandemic, Japan has experienced a significant rise in mortality, with excess deaths surpassing historical projections. Statistical data indicate a sharp increase in mortality rates from 2021 onward, attributed to COVID-19, aging demographics, cardiovascular diseases, and malignancies. Preliminary 2024 data suggest continued excess mortality, fueling public debate. This review analyzes national and municipal mortality trends using official Japanese statistics and comparative data from South Korea, the U.S., and the EU. Findings reveal a sharp mortality rise post-2021 in Japan and South Korea, while Western nations experienced peak deaths in 2020, followed by declines. The review explores contributing factors, including potential vaccine-related adverse effects, declining healthcare access, pandemic-induced stress, and demographic shifts. Notably, older adults’ reluctance to seek medical care led to delayed diagnoses, treatment interruptions, and preventable deaths. Although some argue that declining COVID-19 vaccination rates in 2023 may have contributed to rising mortality in 2024, available data suggest a multifactorial causation. Japan’s rapidly aging population, coupled with increasing mortality and declining birth rates, presents profound social and economic challenges. A nuanced approach, avoiding simplistic causal claims, is crucial for understanding these trends. This review highlights the need for a sustainable societal framework to address demographic shifts and improve healthcare resilience. Future pandemic strategies must balance infection control measures with mitigating unintended health consequences to ensure a more adaptive and effective public health response.

## 1. Introduction

Since the onset of the coronavirus disease (COVID-19) pandemic, rising mortality rates in Japan have become a topic of concern. In particular, reports indicate an increase in excess mortality, defined as the difference between observed deaths and the number expected based on historical data and statistical models [1]. Mortality rates have risen significantly since 2021, posing a major public health challenge. Statistical data up to 2022 suggest that, in addition to COVID-19, factors such as senility, cardiovascular diseases, and malignant neoplasms have contributed to this increase [2]. Moreover, the continued long-term monitoring of excess mortality and other quantitative data beyond 2022 remains essential [3]. In February 2025, nationwide and municipal-level statistical reports on mortality for 2024 were released. Meanwhile, discussions on social media platforms such as X have highlighted growing concerns over the renewed increase in deaths throughout 2024 (Table 1). The increase in mortality during the COVID-19 pandemic has been reported in countries other than Japan [4,5,6,7]. Even after the acute phase of the COVID-19 pandemic subsided, mortality rates in Japan have not decreased. In 2022, they rose again, in 2023 there was a slight decrease or plateau, and in 2024, a trend of rising mortality rates has been observed again.

A recently published short opinion article discusses the relationship between COVID-19 vaccination and excess mortality in Japan [8]. This article describes the sharp increase in excess mortality in Japan following the emergence of the Omicron variant. Hypotheses regarding the potential association between mRNA vaccination and excess mortality include adverse effects such as myocarditis, autoimmune diseases, and cancer, as well as immune suppression due to repeated vaccinations. However, this article does not provide direct evidence demonstrating a causal relationship between mRNA vaccines and excess mortality. Instead, it outlines possible contributing factors and concludes that, due to insufficient research and data, the true cause remains unclear.

To clarify the relationship between vaccination and excess mortality, studies directly investigating the correlation between mRNA COVID-19 vaccination and excess mortality in Japan are necessary. However, as of now, no studies have comprehensively examined this association.

This review first examines trends in mortality counts and crude death rates over the past decade across Japan and its four major cities (the 23 wards of Tokyo, Yokohama, Osaka, and Nagoya), comparing these trends with those observed in other countries. It then explores potential factors contributing to the rise in mortality in Japan since 2021. Finally, the author offers insights into Japan’s future as the country enters an era of unprecedented population decline.

While this review article references numerous online sources, the author wishes to emphasize that these citations are intended solely to confirm objective facts and are not meant to defame or criticize any specific individual or group.

## 2. Development and Efficacy of mRNA Vaccines: A Scientific Assessment of COVID-19 Protection

mRNA COVID-19 vaccines work by delivering mRNA encoding the viral spike protein into the body, prompting cells to produce the antigen and induce an immune response. Their rapid development was enabled by pre-existing mRNA technology research, the prioritization of funding and regulatory processes during the pandemic, and the concurrent execution of large-scale clinical trials. Extensive clinical trials and real-world data have confirmed their high efficacy, particularly in preventing severe disease, as well as their safety, with rare but minimal risks of adverse effects [9].

Japan’s Ministry of Health, Labour, and Welfare, along with related academic societies, strongly recommends additional vaccinations in the fall and winter of 2024 for the elderly and those at high risk of severe disease [10]. They maintain the stance that the benefits of COVID-19 vaccination far outweigh the risks. Numerous studies worldwide support the efficacy of COVID-19 mRNA vaccines, and infectious disease experts continue to recommend vaccination based on these findings.

Compared to unvaccinated individuals, vaccinated individuals have been shown to have a reduced risk of hospitalization and long-term symptoms [11]. In South Korea, vaccinated individuals had lower non-COVID-19 mortality rates compared to unvaccinated individuals [12]. In the Netherlands, the 2023 COVID-19 XBB.1.5 vaccine was associated with reduced hospitalizations and ICU admissions due to COVID-19 [13]. Even in breakthrough infections, vaccination has been shown to suppress household transmission [14]. In Japan, a report indicated that Omicron breakthrough infections in individuals with three or more vaccine doses or prior infections had a lower frequency of systemic symptoms [15]. Vaccination has also been associated with a lower risk of post-acute sequelae of COVID-19, with a third dose further reducing risks of heart failure, arrhythmia, cardiac arrest, pulmonary embolism, and new dialysis compared to two doses [16].

Many studies have reported the beneficial effects of vaccination on long-term symptoms. Booster doses have been shown to mitigate severe disease and post-acute sequelae [17,18]. Pre-infection vaccination has been strongly associated with a reduced risk of being diagnosed with long-term COVID-19 complications [19]. A systematic review and meta-analysis have also confirmed that vaccines reduce post-COVID-19 conditions [20]. Additional reports suggest that the post-COVID-19 syndrome risk is lower among previously vaccinated individuals compared to unvaccinated individuals infected with the Omicron variant [21].

The updated 2023–2024 COVID-19 vaccines have demonstrated protective effects against hospitalizations caused by XBB and JN lineages, providing a rationale for promoting new vaccine formulations [22]. The BNT162b2 XBB vaccine offers statistically significant additional protection against various COVID-19 outcomes. However, older vaccine formulations provided little to no long-term protection against hospitalization, irrespective of the dose number or vaccine type, reinforcing the need for updated vaccinations [23]. The relative effectiveness of ancestral-strain mRNA vaccines against Omicron infections has been shown to be higher than that of viral vector and protein subunit vaccines [24].

Accordingly, the U.S. Advisory Committee on Immunization Practices (ACIP) has recommended COVID-19 vaccination for all individuals aged six months and older using FDA-approved or authorized vaccines for the 2024–2025 season [25].

However, mRNA vaccines have been reported to cause more adverse reactions compared to Novavax’s protein-based COVID-19 vaccine (NVX-CoV2373) [26]. A Japanese study suggested that while COVID-19 mRNA vaccines are generally safe, there was a potential trend of increased pulmonary embolism following the first dose [27].

Despite ongoing discussions regarding COVID-19 vaccination, a substantial proportion of the elderly population in Japan had received booster doses up until 2023. However, the uptake of booster vaccinations in 2024 has declined considerably.

## 3. Analysis of National and Municipal Mortality Trends

Interested in the data circulating on social media, the author first verified the accuracy of annual nationwide mortality trends in Japan over the past decade (since 2015) using confirmed statistics published in the Vital Statistics of Japan by the Ministry of Health, Labour, and Welfare. For 2024, preliminary figures from relevant government websites were used (Table 2).

The line graph and the regression line were created using Microsoft Excel. A line graph illustrating these data is presented in Figure 1A and closely mirrors trends observed on social media. Due to population aging, the annual number of deaths in Japan had been increasing linearly even before the pandemic. A linear regression model was therefore constructed using the number of deaths during the pre-pandemic period (2015–2019) as the baseline. The resulting regression equation is shown in red in the figure. Estimates of excess mortality, calculated as the difference between the actual and predicted total number of deaths (actual − predicted) from 2015 onward, are presented in Figure 1B. Nationwide mortality in Japan showed a slight decline in 2020, the first year of the COVID-19 pandemic, followed by significant increases in 2021 and 2022. The mortality rate remained elevated in 2023 and is projected to persist in 2024.

To assess how these trends compare at the municipal level, line graphs were generated for Japan’s four largest cities by population—Tokyo’s 23 wards, Yokohama, Osaka, and Nagoya—using publicly available statistical data from each municipal government (Table 2 provides the URLs for these data sources). The mortality trends for Tokyo’s 23 wards (Figure 2A), Yokohama City (Figure 2C), Osaka City (Figure 2F), and Nagoya City (Figure 2G) are shown.

Similar to the nationwide trend (Figure 1A), mortality in all four major cities rose sharply from 2021 and remained elevated through 2022 and 2023. Among the cities with preliminary data available for 2024, Yokohama and Osaka recorded further increases. Overall, all four cities exhibited nearly identical mortality trends. Similar to the trend observed for all of Japan, linear regression models were constructed for each of the four major cities—Tokyo 23 wards, Yokohama, Osaka, and Nagoya—using the number of deaths during the pre-pandemic period (2015–2019) as the baseline. The resulting regression equations are shown in red in the figures. In all areas, excess deaths began to increase in 2021, rose sharply in 2022, and, although they slightly decreased in 2023, remained above pre-pandemic levels. Estimates of excess mortality, defined as the difference between the actual total number of deaths and the predicted number of deaths (actual total deaths–predicted total deaths) from 2020 onward, are presented in Figure 2B (Tokyo 23 wards), Figure 2D (Yokohama), Figure 2F (Osaka), and Figure 2H (Nagoya).

Figure 3A shows the trends in crude mortality rates per 100,000 persons over the past decade across Japan and its four major cities. It is evident that the crude mortality rate rose sharply in 2021 across Japan and all these cities and has remained high since 2022. Compared to the national average, major metropolitan areas tend to have a slightly lower aging population. A regression equation was constructed based on crude mortality rates from the pre-pandemic period (2015–2019), as shown in Figure 3A. Figure 3B presents the time trend of the difference between the observed total crude mortality rate and the predicted total crude mortality rate derived from the regression model (i.e., actual minus predicted). Throughout Japan, including the 23 wards of Tokyo, Yokohama City, Osaka City, and Nagoya City, the difference began to increase in 2021, showed a marked rise in 2022, and while the upward trend slowed somewhat after 2023, elevated levels have persisted.

Excluding Osaka, the crude mortality rates in the three major cities are somewhat lower than those of Japan as a whole. In particular, the 23 wards of Tokyo exhibit a lower crude mortality rate, likely due to the influx of younger individuals moving to the city for education and employment, resulting in a lower aging population. Yokohama, which is part of the Tokyo metropolitan area, follows a similar trend, although its crude mortality rate is slightly higher than Tokyo’s. Nagoya still has a higher crude mortality rate.

Osaka’s crude mortality rate is comparable to the national average, likely due to several factors, including a higher aging population than in the Tokyo metropolitan area, significant income disparities, leading to greater health risks and unequal access to medical care among lower-income groups, a higher prevalence of lifestyle-related diseases, such as diabetes, obesity, and smoking, and the impact of living conditions and stress-related factors on health. These factors likely interact in a complex way to contribute to Osaka’s mortality trends. The Gini coefficient, an indicator representing income and wealth inequality, is similar for the Tokyo Metropolis (0.303) and Osaka Prefecture (0.301), while the Kanagawa Prefecture, where Yokohama City is located, has a lower value of 0.267, and the Aichi Prefecture, home to Nagoya City, has a coefficient of 0.286, both lower than Tokyo and Osaka [28]. On the other hand, the Osaka Prefecture continues to show a higher proportion of low-income households compared to the national average, Tokyo, and Aichi Prefecture [29]. For example, in the Tokyo 23 wards, households with an annual income below 3 million yen account for 33% of all households, whereas in Osaka City, this proportion reaches 50%, indicating that despite similar Gini coefficients, Osaka has a larger low-income population [30]. These findings suggest that income-related health risks and disparities in medical access are expanding among low-income groups in Osaka. Health indicators related to lifestyle diseases in Osaka City tend to be higher compared to Tokyo and the national average. The prevalence of diabetes among males in Osaka City is 17.6%, and among females it is 7.9%, both above the national average. The obesity rate is also higher, with 35.2% for males and 18.7% for females. Furthermore, the smoking rate in Osaka City is 17.7%, ranking third highest among Japan’s designated cities after Kitakyushu and Sapporo [31,32].

Figure 4A illustrates the trends in all-cause age-standardized mortality rates (ASMRs) across Japan. In Japan, ASMRs for both men and women have shown a long-term downward trend. This decline is likely attributable to a combination of factors, including advancements in medical technology and healthcare systems, improvements in public health and social infrastructure, the development of welfare and long-term care systems for the elderly, as well as increased public health awareness and healthier lifestyles. These achievements can be seen as the cumulative result of sustained efforts by the Japanese population over many years. However, since 2021, ASMRs have reversed course and begun to increase in both sexes. To investigate this further, the author utilized an independent website that draws on official statistics published by public institutions [33] to examine mortality trends. Figure 4B presents the trends in crude (non-age-standardized) mortality counts across Japan, which mirrors the pattern seen in Figure 1A. Figure 4C shows the trends in age-standardized mortality counts, which, like the ASMRs in Figure 4A, had been decreasing until 2020 but then began to rise after 2021. To assess whether this post-2021 increase in age-standardized mortality can be explained solely by population aging, the author further analyzed age-standardized mortality counts specifically for individuals under the age of 20, as shown in Figure 4D. Surprisingly, even among minors, age-standardized mortality counts have been increasing since 2021, suggesting that the recent rise in mortality cannot be explained by aging alone.

## 4. Mortality Trends Outside Japan

How have mortality trends evolved in countries outside Japan? To investigate this, population statistics from South Korea [34] were collected and analyzed using the same methodology. The results revealed a sharp increase in mortality beginning in 2022, followed by persistently high mortality in 2023, demonstrating a pattern similar to that observed in Japan (Figure 5).

Using data from Our World in Data [35], the author plotted mortality trends for Japan (Figure 6A) and South Korea (Figure 6B). While minor discrepancies exist due to differences in data aggregation methods, the overall trends align with those in Figure 1A and Figure 5. Specifically, no significant increase in mortality was observed in 2020; however, mortality began rising from 2021 to 2022. In contrast, mortality trends in the European Union (Figure 6C) and the United States (Figure 6D) followed a different trajectory, with both regions experiencing a sharp increase in deaths during the first year of the COVID-19 pandemic (2020), followed by a decline in mortality from 2022 to 2023.

During the early phase of the pandemic, East Asian countries such as Japan and South Korea recorded significantly fewer severe cases and deaths from COVID-19 compared with Western nations.

The underlying reasons for this disparity remained unclear and were often referred to as “Factor X”. However, leading infectious disease experts categorically dismissed the existence of “Factor X”, considering it an illusion. The author has repeatedly questioned whether this dismissal was appropriate [10,36]. Several hypotheses have been proposed to explain “Factor X”, which refers to the relatively low rates of severe illness and mortality due to COVID-19 in certain countries, including Japan. One such hypothesis suggests that countries with mandatory BCG vaccination policies exhibited fewer severe cases and deaths [37]. Another proposes that differences in human leukocyte antigen (HLA) types among East Asian populations, including the Japanese, may influence innate immunity and T-cell responses to the virus [38]. A third hypothesis posits that racial or ethnic differences in the expression levels of ACE2—the cellular receptor utilized by SARS-CoV-2 for entry—could account for varying susceptibility to infection [39]. In addition, behavioral and cultural practices common in Japan, such as mask-wearing, rigorous hygiene habits, and the custom of removing shoes indoors, have been considered potential contributing factors to the limited spread of infection.

Given that the impact of infectious diseases can vary significantly depending on genetic, environmental, and regional factors, it is crucial to account for these differences when formulating future pandemic response strategies. It should be noted that much of the positive evidence regarding COVID-19 vaccines was generated in the early phase of their introduction (2021–2022) and primarily focused on populations in Western countries [9,40]. Therefore, applying such evidence directly to the current situation in East Asia may not be appropriate.

## 5. Excess Mortality and COVID-19 Vaccination

Preliminary mortality data for each municipality in January 2025 were released in February 2025, revealing a significant increase in deaths compared to the previous year. This sharp rise has become a topic of widespread discussion (Table 1).

A study in the United Kingdom has reported that the standardized mortality ratio (SMR) for all-cause mortality is higher in COVID-19 vaccinated individuals compared to unvaccinated individuals [41]. Opponents of COVID-19 vaccination argue that this surge in excess mortality is a direct consequence of the harmful effects of COVID-19 vaccines [42,43]. Infectious disease experts promoting COVID-19 vaccination appear in newspaper advertisements sponsored by vaccine manufacturers, continuing to advocate for booster doses [10]. However, it has been pointed out that the number of cases approved for compensation under the COVID-19 vaccines injury relief system, including deaths, far exceeds the total for all other vaccines combined over the past more than 40 years [44]. The number of deaths officially acknowledged under the Immunization Health Damage Relief System approaches 1000. It is anticipated that some may argue that the high number of compensation cases is due to the overwhelmingly large number of COVID-19 vaccines recipients. However, it has also been claimed that the incidence of adverse events is significantly higher for COVID-19 vaccines compared to influenza vaccines, despite both being widely administered [45].

Given such reports, it is understandable that the general public may grow increasingly distrustful of COVID-19 vaccines.

While the possibility that long-term effects of mRNA vaccines may have contributed to increased rates of cardiovascular diseases or immune dysfunction cannot be ruled out, multiple factors need to be considered [8]. After vaccination, cases of herpesvirus [46], varicella-zoster virus infections [47], autoimmune diseases [48,49], thrombotic thrombocytopenia [50], nephrosis [51,52], nephritis [53,54], myocarditis, and pericarditis [55,56,57,58,59,60] have been reported. Notably, a significant rise in deaths was already evident in 2021, the year COVID-19 vaccinations began. Some researchers have speculated that the increase in mortality observed from 2024 to 2025 could be a delayed manifestation of vaccine-related adverse effects or long-term complications.

## 6. Other Contributing Factors to Increased Mortality

However, in addition to potential vaccine-related impacts, several other factors may have contributed to the rise in mortality. A key demographic factor is Japan’s aging population—by 2024, the baby boomer generation will have reached 75 years and older, an age range associated with naturally rising mortality rates. Healthcare access issues may have also played a role, including increased strain on medical institutions and a decline in healthcare-seeking behaviors. Japanese older adults have traditionally sought medical care more frequently than those in other countries. However, following the COVID-19 pandemic, an increasing number of elderly individuals refrained from seeking medical attention. As a result, cases emerged in which the treatment of previously managed diseases was interrupted, regular screenings were not conducted, making early detection difficult, and, in some instances, patients were unable to access appropriate medical care after disease onset, leading to preventable deaths. These effects may still persist today. Furthermore, this reluctance to seek medical care likely contributed to insufficient prevention and early intervention for conditions such as cancer, cardiovascular diseases, and cerebrovascular diseases.

The implementation of COVID-19 countermeasures aimed at minimizing interpersonal contact led to a decline in routine health check-ups among older adults, which is likely to have had a direct causal relationship with the observed increase in overall mortality. During the pandemic, elderly individuals increasingly refrained from seeking medical care due to concerns about infection risk. As a result, reductions in cancer screening participation and increases in the self-discontinuation of prescribed medications were observed. In a survey asking about “diseases for which patients were unable to seek care due to the impact of COVID-19”, 59.3% of care managers in Tokyo responded affirmatively. Among these, the most frequently cited conditions were “lifestyle-related diseases” (27.9%), followed by “dementia” (23.3%) and “dental diseases” (19.8%) [61]. These trends suggest that the health management of older adults and the healthcare delivery system were adversely affected during the pandemic, highlighting the need for targeted interventions going forward.

The adverse effects of delayed diagnosis and treatment have also been observed in patients with cardiovascular diseases requiring emergency care. Alarmist messaging by some infectious disease experts contributed to heightened public anxiety, which in turn led to delays in the diagnosis and treatment of more severe diseases. In Japan, the COVID-19 pandemic had a notable impact on patients with ST-elevation myocardial infarction (STEMI). Following the COVID-19 pandemic, primary PCI was performed significantly less frequently, and the incidence of mechanical complications resulting from ST-elevation myocardial infarction (STEMI) increased. Failing to seek immediate medical attention and waiting at home when experiencing heart attack symptoms may worsen outcomes for patients with STEMI [62]. A decrease in hospital admissions, delays in seeking care, and treatment initiation were observed [63]. The door-to-balloon time (DTBT), a critical indicator of timely reperfusion through primary percutaneous coronary intervention (PCI), was significantly prolonged during the pandemic [64,65].

Reperfusion therapy for acute ischemic stroke was also impacted by the COVID-19 pandemic. During the state of emergency, the number of stroke admissions decreased, and the time from hospital arrival to imaging and thrombolysis was longer compared to the period before the pandemic [66]. Thus, the fear of contracting COVID-19 has led to delays in the treatment of other life-threatening emergency conditions.

Furthermore, the pandemic was associated with a reduction in cancer screening and early detection. In Japan, population-based cancer screening participation declined by approximately 10–30% during the pandemic [67]. As of 2024, although the number of cancer screenings showed a partial recovery in 2021, levels have not returned to pre-pandemic baselines, except for endoscopic gastric cancer screening. This continued decline raises concerns about potential delays in cancer diagnosis and increases in cancer-related mortality, warranting close monitoring [68]. Notably, a transient decrease in the proportion of early-stage colorectal cancer diagnoses was also observed following the declaration of a state of emergency. Although some reports noted that treatment outcomes at individual medical institutions were not necessarily worsened, the cumulative impact of widespread disruption to the healthcare system likely contributed to adverse effects at the population level.

Moreover, frailty caused by the COVID-19 pandemic is also a problem due to older adults being afraid of contracting COVID-19, which has caused them to refrain excessively and stay at home. The aftermath of the pandemic has led to heightened stress levels and reduced routine health check-ups, potentially exacerbating chronic diseases.

It is well established that physical activity enhances antiviral immunity and may serve as a countermeasure against immune senescence [69]. Conversely, it has been suggested that reduced physical activity due to self-isolation and other infection control measures may have the opposite effect [70].

Several reports from Japan have examined the relationship between the COVID-19 pandemic and frailty and sarcopenia in older adults. In the first year of the pandemic, both physical activity and social engagement significantly declined, as indicated by previous studies [71,72]. The prevalence of frailty among older adults steadily increased from the pre-pandemic period through the first and second years of the pandemic [73]. A retrospective cohort study found that behavioral restrictions during the pandemic contributed to the loss of skeletal muscle mass in older patients with type 2 diabetes. These findings underscore the importance of interventions, such as exercise promotion and adequate nutritional intake, to prevent muscle loss [74]. Additionally, both older men and women experienced a decline in trunk muscle mass during the pandemic, suggesting the progression of sarcopenia. Muscle loss was also associated with increased risks of falls and cognitive decline in older men [75]. It has also been reported that healthy and active older adults require specific strategies to maintain trunk muscle mass in order to prevent so-called “corona-frailty” [76]. Furthermore, corticosteroid use in the treatment of COVID-19 has been identified as a risk factor for the development of osteonecrosis of the femoral head [77] and may also contribute to the onset of diabetes [78]. These findings suggest a potential link between COVID-19-related changes and the increased risk of cardiovascular events and mid-term mortality.

The reasons why Japanese older adults refrained from visiting doctors are believed to be largely influenced by the fear of infection risk, especially due to the widespread public health campaign during the early days of the 2020 pandemic. Following the recommendation of Professor Nishiura of Hokkaido University, who was a member of the Cluster Response Team at the Ministry of Health, Labour, and Welfare at the time, there was a major push to reduce contact with others by at least 70%, and as much as possible by 80%. Other factors, such as overcrowding in medical facilities, mental stress, and the desire to avoid placing additional burdens on the healthcare system, may have also contributed.

The author has previously highlighted concerns that prioritizing infection control may have delayed the diagnosis of other diseases [36], and that strict patient isolation protocols may have hindered flexible responses in clinical settings, potentially lowering the overall quality of medical care [79]. These can be considered indirect adverse effects of COVID-19 infection control measures.

During the COVID-19 pandemic, social and psychological stress was exemplified by symbolic incidents of discrimination and prejudice against healthcare workers. Heightened fear and uncertainty surrounding infection led, in some instances, to a perception of healthcare professionals as potential sources of contagion. In Japan, several emblematic cases of such discrimination have been reported. These include children of healthcare workers being denied access to daycare centers and schools, healthcare professionals being refused taxi rides, and even being pressured by neighbors to move out of their residences. Such experiences caused significant psychological distress among healthcare workers, including symptoms consistent with post-traumatic stress disorder (PTSD), and had substantial social repercussions [80]. Another symbolic and socially significant issue was the imposition of excessive visitation restrictions in the name of infection control. These restrictions often exacerbated anxiety and distress among patients’ families, who increasingly required information and emotional support from medical staff. In neonatal intensive care units (NICUs), families reported disruptions in bonding with their infants and disturbances in family dynamics. For healthcare providers, visitation restrictions introduced ethical dilemmas and increased burdens related to communication and provision of social support [81].

While some may argue that the absence of infection control measures would have led to a higher number of direct COVID-19 deaths, it is deeply ironic that the extensive efforts, financial costs, and sacrifices made for infection control may have ultimately contributed to increased stress levels, a decline in routine health checkups, the progression of frailty in older adults, and an overall rise in mortality.

## 7. Non-Communicable Disease Mortality Trends and the Role of Hybrid Immunity in Japan

Trends in mortality due to non-communicable diseases (NCDs) were also examined. In Japan, age-standardized mortality rates for heart disease had been declining up to 2020, in parallel with overall all-cause age-standardized mortality rates. However, this trend reversed after 2021, with a noticeable increase observed thereafter (Figure 7A). When examining age-standardized mortality from acute cardiovascular diseases using data obtained from an independent website, both acute myocardial infarction (Figure 7B) and cerebrovascular disease (Figure 7C) exhibited a consistent decline up to 2020. However, beginning in 2021, these trends plateaued, suggesting a deviation from the previous downward trajectory.

Age-standardized mortality from all malignant neoplasms has shown a long-term decreasing trend, and this trend has not clearly reversed since the onset of the pandemic (Figure 8A–D). However, data obtained from an independent website suggest a pandemic-associated increase in specific types of cancer, including breast cancer in women (Figure 8E), ovarian cancer (Figure 8F), and leukemia (Figure 8G). Specifically, between 2021 and 2022, breast cancer mortality increased by 6.1%, colorectal cancer by 1.2%, uterine cancer by 4.2%, and pancreatic cancer by 1.7%, suggesting that while overall cancer mortality may not have increased, certain cancer types—especially among women—have shown rising trends. Continuous monitoring is warranted to assess whether these patterns persist [2].

Age-standardized mortality due to senility had already been on an upward trend prior to the pandemic, largely attributable to population aging. However, this increase became even more pronounced after 2021 (Figure 9A,B).

Conversely, proponents of COVID-19 vaccination may argue that the rise in mortality observed in 2024 is linked to a decline in vaccine uptake in 2023. A report suggested that COVID-19 vaccination contributes to reducing excess mortality among the elderly [82]. Furthermore, the VENUS study, led by Kyushu University, suggests that booster doses of mRNA vaccines do not increase the risk of mortality [83]. However, it remains unclear whether they contribute to a reduction in mortality risk. Proponents assert that individuals—particularly older adults and those with underlying health conditions—face an increased risk of severe disease and death due to decreased vaccine coverage.

Given the substantial decline in COVID-19 severity by 2023, it remains challenging to definitively attribute excess mortality primarily to reduced vaccination rates. Although recent estimates indicate approximately 30,000 COVID-19-related deaths annually [84], this figure alone does not fully explain the observed increase in excess mortality. The transition from comprehensive to alternative reporting systems may have resulted in underreporting, while earlier systems may have overestimated mortality by including cases in which COVID-19 was not the primary cause of death. Such uncertainty complicates efforts to accurately assess the specific contribution of COVID-19 to recent excess mortality trends.

When considering the demographic impact of COVID-19 itself, it is essential to take into account the effects of herd immunity. In Japan, the majority of the population has received at least two doses of an mRNA vaccine, followed by natural infection, leading to a high prevalence of hybrid immunity. The author has previously reported that experiencing both natural infection with the virus and mRNA vaccination leads to a significant increase in antibody titers due to hybrid immunity, with antibody levels being maintained at a high level for an extended period (over one year) [85,86,87]. It is suggested that when antibody titers remain elevated, the effectiveness in preventing infection is sustained, and the risk of severe disease is reduced. In other words, even if only the initial two doses of vaccination are administered, followed by natural infection, it can be expected that the risk of severe illness from COVID-19 would be extremely low in healthy individuals.

To avoid any misunderstanding, the author clarifies that he is not entirely skeptical of COVID-19 vaccines and considers them an exceptionally effective technology for these reasons. However, the author cannot help but feel discomfort with the outright denial of the negative aspects of the COVID-19 vaccine or the dismissive attitude toward individuals who express skepticism about vaccination. It is entirely reasonable that some individuals initially had concerns regarding the efficacy and safety of mRNA vaccines, as they represented a novel technology and were developed within an exceptionally short timeframe.

## 8. The Complexity of Mortality Trends and the Need for Careful Analysis

It is difficult to definitively determine whether the arguments presented by either proponents or opponents of COVID-19 vaccination are simply “correct” or “incorrect”. Changes in mortality rates are influenced by multiple factors, necessitating careful and rigorous data analysis. Rather than drawing simplistic causal conclusions, it is crucial to continue examining this issue from multiple perspectives. When COVID-19 vaccinations first started, some doctors appeared on social media and other platforms, tirelessly emphasizing the safety of the vaccine [88]. Some proponents of COVID-19 vaccination expressed extreme, aggressive, and emotionally charged opinions, appearing intolerant of any negative views on the vaccines [89,90]. However, they likely believed in the effectiveness of vaccination and were acting in the best interest of public health with no ill intent.

Regardless of one’s stance, merely criticizing those with opposing views by claiming that their arguments lack evidence or scientific basis may backfire, as the same argument can be directed in return, ultimately undermining one’s own position. It is desirable for both sides to remain objective and avoid extreme claims, such as “vaccines are absolutely effective and safe” or “vaccines are the sole cause”. A cautious and rational approach is essential, emphasizing the careful evaluation of data and distinguishing correlation from causation.

Debates over COVID-19 vaccines and infection control measures are deeply polarized, often culminating in irreconcilable disputes that highlight divisions rather than fostering meaningful dialogue. While opposing sides frequently dismiss each other’s claims, it is crucial to acknowledge the coexistence of differing perspectives and advocate for policies that respect individual choice. Both proponents and critics present valid arguments, yet their discourse is sometimes marked by emotional intensity, aggression, and overreach, raising concerns about the objectivity of their positions. Additionally, both sides tend to selectively disseminate information, emphasizing points that best support their respective viewpoints.

To be honest, the author was initially skeptical about the accuracy of social media posts that reported a significant increase in mortality by the end of February 2025. However, as demonstrated in this study, data extracted from publicly accessible sources, including the Ministry of Health, Labour, and Welfare and various municipal governments, and plotted on a simple line graph, confirmed that the reported trend was indeed real. Readers interested in verifying this finding are encouraged to consult the URLs listed in Table 2, which are linked to the statistical data from each municipality. Those who identify any discrepancies or potential errors in the data are welcome to contact the author.

In the modern era, information disseminated via social networking services (SNSs) on the internet holds significant influence alongside official statements issued by governmental authorities and academic societies. While such information varies in quality—some may be misleading and potentially detrimental to public decision-making—insightful and sharp analyses are also frequently observed. It is anticipated that the incorporation of such perspectives into policy-making will become increasingly important in the future.

In general, information disseminated via social media is regarded as having limited evidentiary value and is therefore typically excluded from the academic literature. However, during the COVID-19 pandemic, viewpoints that deviated from the prevailing consensus—particularly those that raised concerns regarding infection control measures or vaccination campaigns promoted by governmental and academic authorities—were often met with social stigma or were labeled as misinformation, regardless of their content or intent.

As discussed in this manuscript, the increase in all-cause mortality observed in Japan after the pandemic cannot be sufficiently explained by population aging or the direct impact of COVID-19 alone. While alternative explanations remain underexplored in the academic literature, potential contributing factors such as the unintended consequences of prolonged infection control measures and possible adverse reactions to vaccination warrant careful and impartial investigation. Currently, peer-reviewed studies explicitly addressing these factors remain extremely limited.

One example of recent efforts to engage with public discourse outside conventional academic sources includes a Japanese publication that systematically analyzes public concerns expressed on social media regarding pandemic-related health outcomes [91]. Although such sources are not substitutes for empirical evidence, they can offer insight into neglected perspectives that may merit further formal investigation.

It is also noteworthy that several claims initially dismissed as misinformation on social media have, in retrospect, been partially reconsidered. For instance, speculation that SARS-CoV-2 might have originated from a laboratory was widely circulated on social media early in the pandemic. In February 2020, a group of prominent public health experts published a statement in *The Lancet* categorically rejecting such claims as conspiracy theories [92]. However, in April 2025, a report released by the U.S. government reignited debate by suggesting that SARS-CoV-2 may have originated from a laboratory in Wuhan, based on specific genomic characteristics not typically found in nature [93]. Although the origin of SARS-CoV-2 lies beyond the scope of this paper, this case illustrates how hypotheses that were once marginalized may, over time, undergo formal re-evaluation by official institutions.

In light of these considerations, the inclusion of references to carefully selected social media-derived content in this manuscript is intended not as an endorsement of unverified claims, but rather as a reflection of the evolving nature of scientific discourse and the need to remain open to revisiting earlier assumptions as new evidence emerges.

## 9. Examining Japan’s Rising Mortality Rate: An International Comparison and Contributing Factors

Based on our previous considerations, the author aims to explore the reasons for the notable increase in Japan’s mortality rate from an international perspective, particularly in comparison with Western countries.

One possible explanation is Japan’s super-aged society. Japan has a higher proportion of elderly individuals than other countries, which naturally contributes to a greater prevalence of underlying health conditions, potentially leading to an increased mortality rate.

A key factor is the direct impact of COVID-19 itself. In Japan, there was a significant increase in COVID-19-related deaths among the elderly, particularly from the latter half of 2022 onward. In contrast, Western countries experienced their peak infections between 2020 and 2021, suggesting that Japan’s peak was delayed.

Another consideration is the potential contribution of adverse events related to COVID-19 vaccination, particularly thrombotic events and cardiovascular diseases, to increased mortality. While vaccination rates declined in Western countries after 2022, Japan continued booster vaccinations, especially among the elderly, with many receiving six to seven doses. Additionally, mRNA vaccines were predominantly used in Japan, whereas other countries also incorporated protein-based vaccines. Notably, the Novavax protein-based COVID-19 vaccine (NVX-CoV2373) has been reported to have fewer side effects compared to mRNA vaccines [26]. To assess the impact of vaccines, an urgent investigation is needed to analyze the correlation between Japan’s excess mortality rate and vaccination rate since 2022.

The impact of COVID-19 policies should also be considered. Japan implemented strict infection control measures—including mask mandates, restrictions on outings, and behavioral limitations—for an extended period. This prolonged adherence to restrictive measures may have contributed to increased frailty and immune decline among the elderly. In contrast, Western countries lifted restrictions earlier, allowing for natural exposure to the virus and potentially facilitating immunity acquisition.

Healthcare and social system factors may also have played a role. In Japan, the excessive emphasis on infection control may have overwhelmed medical institutions, leading to delays in the diagnosis and treatment of cancer and chronic diseases. Given that the peak of COVID-19 infections in Western countries occurred earlier, their healthcare systems may have recovered sooner. Furthermore, in Japan, the strain on medical resources resulted in increased cases of emergency transport difficulties, delaying treatment for myocardial infarction and stroke. In other countries, emergency medical services unrelated to COVID-19 may have resumed more promptly.

Lastly, the prolonged implementation of strict infection control measures in Japan had significant economic and social repercussions. The resulting economic downturn, rising inflation, and stagnation of income growth likely deteriorated the quality of life (QOL) for many individuals. Notably, the suicide rate increased, particularly among young adults and middle-aged men. In contrast, other countries experienced swifter economic recovery and the resumption of social activities.

Regarding suicide, the increase in suicide rates since 2020 has emerged as a significant social issue in Japan, with a particularly pronounced impact on women. During the second wave of the pandemic (July–October 2020), the monthly suicide rate increased by 16%, with a sharper rise among women (37%) and children and adolescents (49%) [94]. Although the rise in female suicides in Japan during the pandemic has been attributed to factors such as worsening economic conditions, financial insecurity, and social isolation, studies have shown that the increase occurred across various occupations, motives, and age groups [95]. Among men, suicide rates increased in 2022 based on age-standardized mortality rates (ASMRs), marking the first increase since 2009, following the Lehman shock (Figure 10A). In particular, suicide rates were higher in 2022 among males aged 50–59 and those aged 80 and above [96]. Data obtained from an independent website on age-standardized suicide mortality further support this trend, demonstrating an increase in suicide rates among women in 2020 (Figure 10C) and among men in 2022 (Figure 10B), consistent with the patterns observed in Figure 10A. When considering the overall population (Figure 10D), the previously observed declining trend in age-standardized suicide mortality has ceased following the onset of the pandemic. Notably, there has been a marked acceleration in the increase in age-standardized mortality due to suicide among individuals under 20 years old since the beginning of the pandemic (Figure 10E). This trend is consistent with the increase in all-cause age-standardized mortality rates observed among minors (Figure 4D), suggesting the need for further investigation into the social and psychological challenges faced by children and adolescents.

## 10. Japan’s Declining Population and Its Implications

Although deaths in Japan have risen significantly, the birth rate has sharply declined. Japan has entered an era of extremely low birth rates and increasing mortality. Since the onset of the COVID-19 pandemic, mortality rates have exceeded initial projections, highlighting a more severe population decline than previously anticipated. Measures such as providing support for child-rearing and considering the acceptance of immigrants are being explored to address the declining birthrate, but their effectiveness is likely to be limited. This demographic shift poses substantial challenges, including economic stagnation, diminished international influence, difficulties in maintaining social infrastructure and welfare systems, and security concerns. These issues have profound implications for Japanese society. Whether or not we choose to acknowledge this reality, we must confront it objectively. Moving forward, the key challenge is to develop a sustainable society based on a model of shrinking equilibrium—one that accommodates population decline without economic and social collapse, while integrating economic and social planning with demographic trends.

In Japan, where the majority of food and energy supplies rely on imports, some argue that maintaining a population of 120 million will become increasingly difficult as the country’s economic contraction and decline are anticipated in the future, potentially leading to a loss of the financial capacity to sustain such imports. Japan is the first major nation in modern history to experience such a rapid population decline. Low birth rates and an aging population make demographic decline a pressing issue not only in Japan but also in other developed countries. In contrast, many developing nations continue to struggle with rapid population growth. The global challenges of food and energy crises remain pressing, and population decline may help reduce environmental burdens. From an environmental conservation perspective, a shrinking population is not necessarily an entirely negative phenomenon. By pioneering a model for a society adapting to population decline, Japan can offer valuable insights for building a sustainable future.

## 11. Conclusions

The rising mortality in Japan, particularly since the onset of the COVID-19 pandemic, represent a significant public health concern. The analysis of nationwide and municipal mortality trends highlights a clear pattern of increased deaths beginning in 2021, which persisted through 2023 and shows projections for continued elevation in 2024. While COVID-19, alongside factors like cardiovascular diseases and cancer, has undoubtedly played a role, other complex factors, including the aging population, healthcare access issues, and possible long-term effects of COVID-19 vaccinations, must also be considered.

There are some limitations in this review article. The author was unable to calculate age-standardized mortality rates due to a lack of access to age-stratified mortality data. Consequently, it was necessary to rely on previously published studies and make secondary use of figures from an independent website that draws on official statistics published by public institutions. It is also possible that age-stratified mortality data for 2024 have not yet been released. The continued monitoring of age-standardized mortality rates beyond 2025 will be important. It is necessary to conduct further research, including cohort studies comparing mortality rates between vaccinated and unvaccinated groups to demonstrate a true causal relationship between COVID-19 vaccination and mortality, studies comparing international excess mortality data and examining correlations with vaccination rates, and research analyzing differences in mortality rates based on the presence or absence of booster vaccinations.

Moreover, increases in mortality are likely influenced by multiple factors, including age, vaccination status, access to healthcare, social security systems, and life expectancy. Future studies should employ multivariable analytical approaches that take these contributing factors into account.

The challenges posed by Japan’s declining birthrate and aging population further complicate the situation, necessitating the careful examination of the interconnected demographic and public health issues.

Despite the ongoing debate surrounding the causes of increased mortality, it is essential to adopt a balanced, data-driven approach to understanding the factors at play. A cautious and objective analysis of the available data, without resorting to extreme positions on either side of the vaccination debate, is crucial for developing appropriate public health responses. Japan’s demographic decline, while presenting significant challenges, also offers an opportunity for the country to pioneer sustainable solutions for managing population change, potentially providing valuable insights for other nations facing similar issues. Moving forward, Japan must confront these demographic and public health challenges with strategic, evidence-based planning to ensure a stable and sustainable future.

## Figures and Tables

**Figure 1 healthcare-13-01305-f001:**
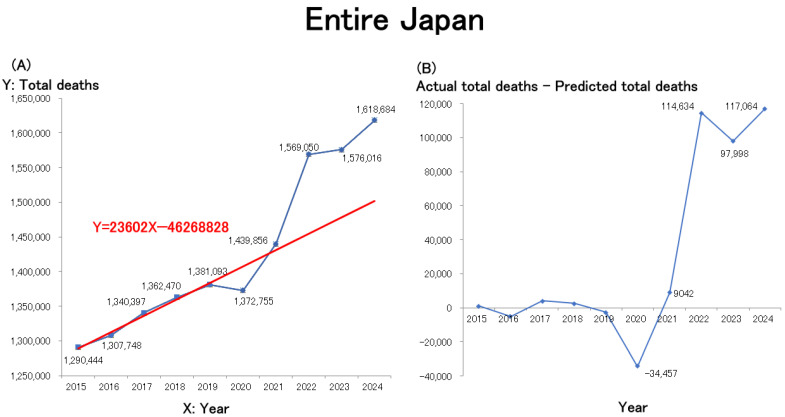
(**A**): Trends in the annual number of deaths in Japan since 2015. X: Year is in the Gregorian calendar. The regression equation and the corresponding linear regression line for the pre-pandemic period (2015–2019) are shown in red in the figure. (**B**): Trends in the annual difference between the actual and predicted total number of deaths (i.e., actual minus predicted).

**Figure 2 healthcare-13-01305-f002:**
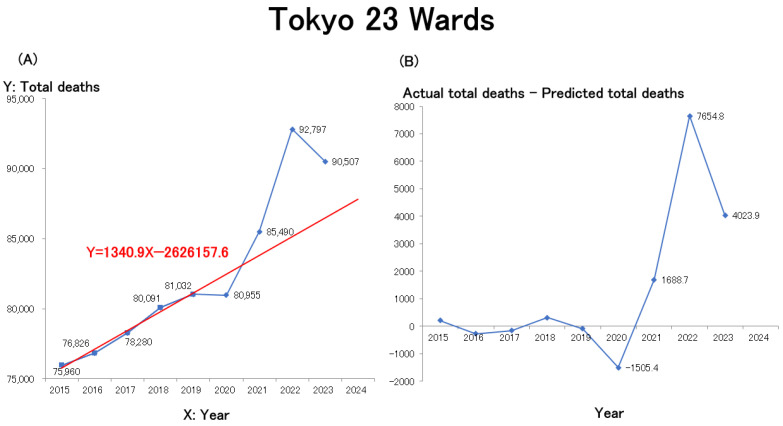
Trends in the annual number of deaths in Japan’s four most populous cities since 2015. (**A**): Tokyo 23 wards, (**C**): Yokohama, (**E**): Osaka, (**G**): Nagoya. X: Year is in the Gregorian calendar. The regression equations and corresponding linear regression lines for the pre-pandemic period (2015–2019) are shown in red in each figure. Trends in annual estimates of excess mortality (actual total deaths minus predicted total deaths) in Japan’s four most populous cities since 2015: (**B**) Tokyo 23 wards, (**D**) Yokohama, (**F**) Osaka, and (**H**) Nagoya.

**Figure 3 healthcare-13-01305-f003:**
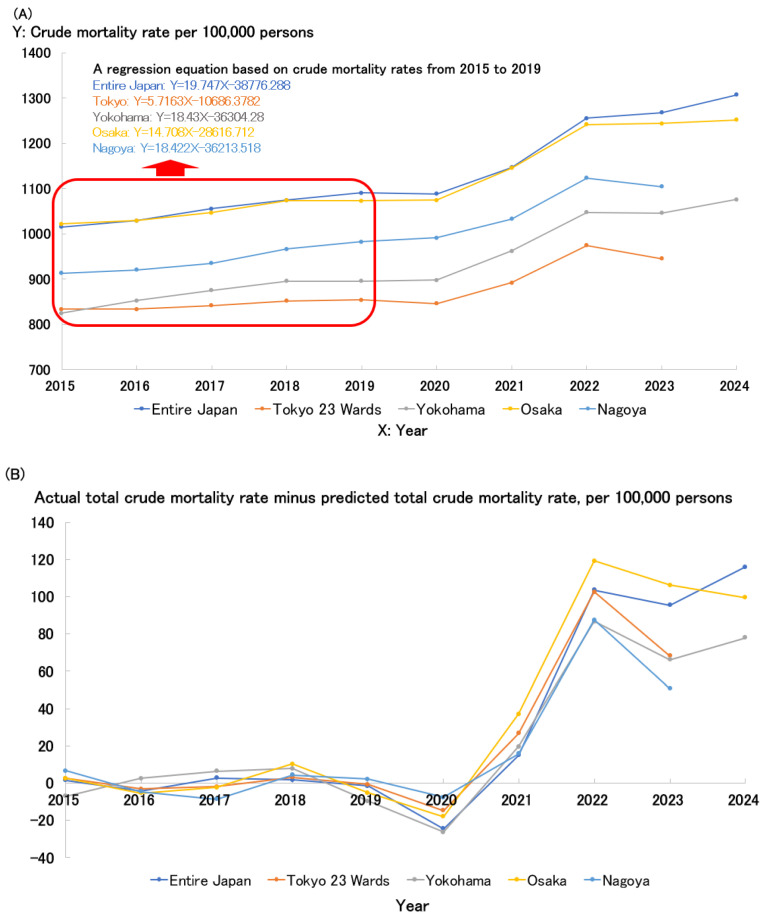
(**A**): Crude mortality rate per 100,000 persons (2015–2024). X: Year is in the Gregorian calendar. (**B**): The time trend of the difference between the observed total crude mortality rate and the predicted total crude mortality rate derived from the regression model (i.e., observed minus predicted).

**Figure 4 healthcare-13-01305-f004:**
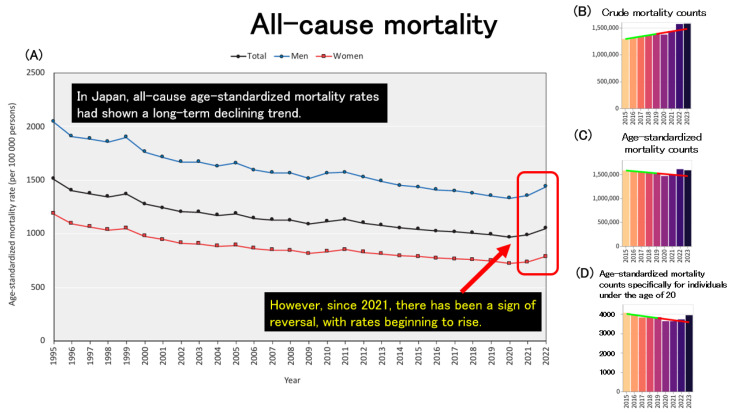
(**A**): Trends in all-cause age-standardized mortality rates (ASMRs) across Japan [2]. (**B**): Trends in crude (non-age-standardized) all-cause mortality counts in both sexes. (**C**): Trends in age-standardized all-cause mortality counts in both sexes. (**D**): Trends in age-standardized all-cause mortality counts among individuals under 20 years of age, in both sexes. (**B**–**D**): The regression line was constructed based on data from the period up to 2019 [33].

**Figure 5 healthcare-13-01305-f005:**
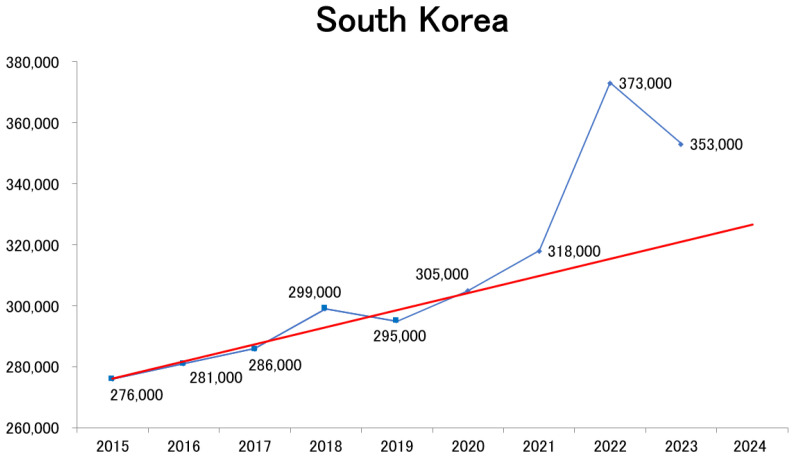
Trends in the annual number of deaths in South Korea since 2015 [34]. The linear regression line for the pre-pandemic period (2015–2019) is shown in red in the figure.

**Figure 6 healthcare-13-01305-f006:**
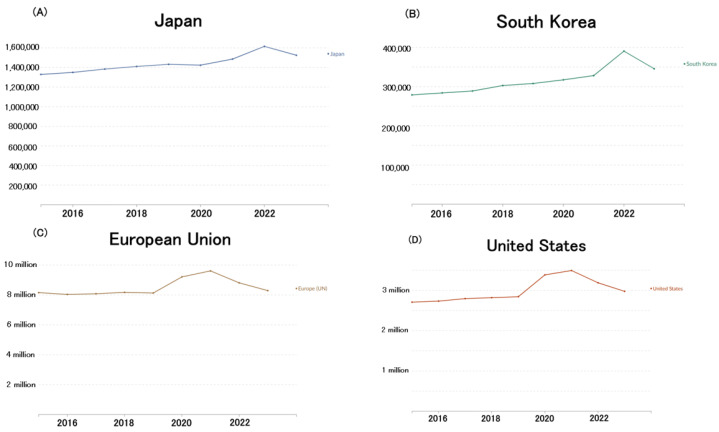
Trends in the number of deaths in each country since 2015. (**A**): Japan; (**B**): South Korea; (**C**): European Union; and (**D**): United States [35].

**Figure 7 healthcare-13-01305-f007:**
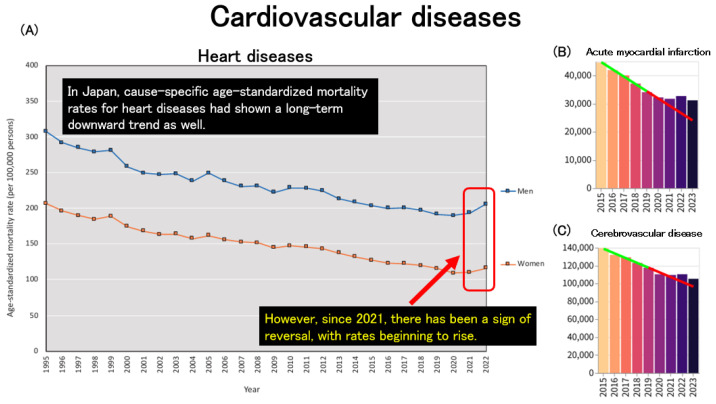
Trends in age-standardized mortality due to cardiovascular diseases. (**A**): Trends in age-standardized mortality rates for heart disease across Japan [2]. (**B**): Trends in age-standardized mortality counts due to acute myocardial infarction in both sexes. (**C**): Trends in age-standardized mortality counts due to cerebrovascular disease in both sexes. (**B**,**C**): The regression line was constructed based on data from the period up to 2019 [33].

**Figure 8 healthcare-13-01305-f008:**
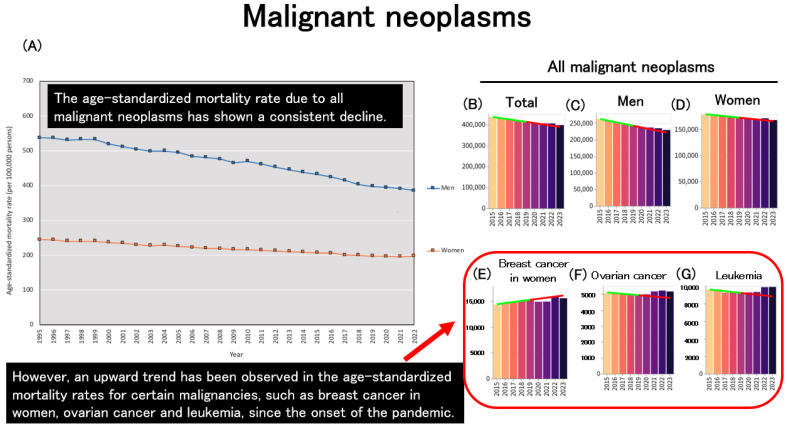
Trends in age-standardized mortality due to malignant neoplasms. (**A**): Age-standardized mortality rates for all malignant neoplasms across Japan [2]. (**B**): Age-standardized mortality counts for all malignant neoplasms in both sexes. (**C**): Age-standardized mortality counts for all malignant neoplasms among men. (**D**): Age-standardized mortality counts for all malignant neoplasms among women. (**E**): Age-standardized mortality counts due to breast cancer among women. (**F**): Age-standardized mortality counts due to ovarian cancer among women. (**G**): Age-standardized mortality counts due to leukemia in both sexes. (**B**–**G**): The regression line was constructed based on data from the period up to 2019 [33].

**Figure 9 healthcare-13-01305-f009:**
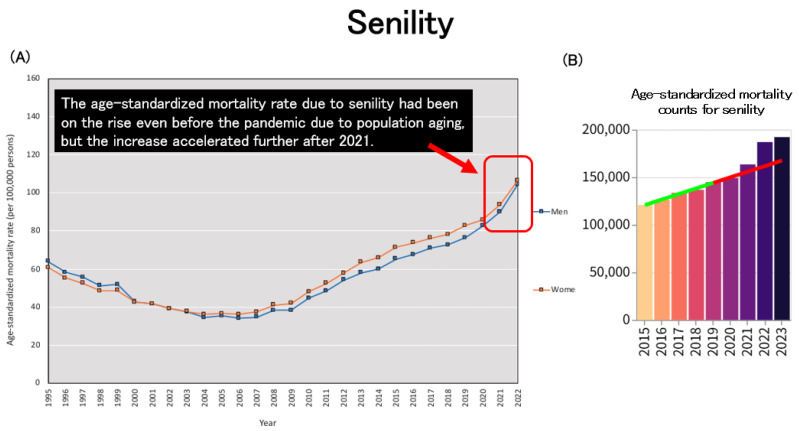
Trends in age-standardized mortality due to senility. (**A**): Age-standardized mortality rates for senility across Japan [2]. (**B**): Age-standardized mortality counts for senility in both sexes. (**B**): The regression line was constructed based on data from the period up to 2019 [33].

**Figure 10 healthcare-13-01305-f010:**
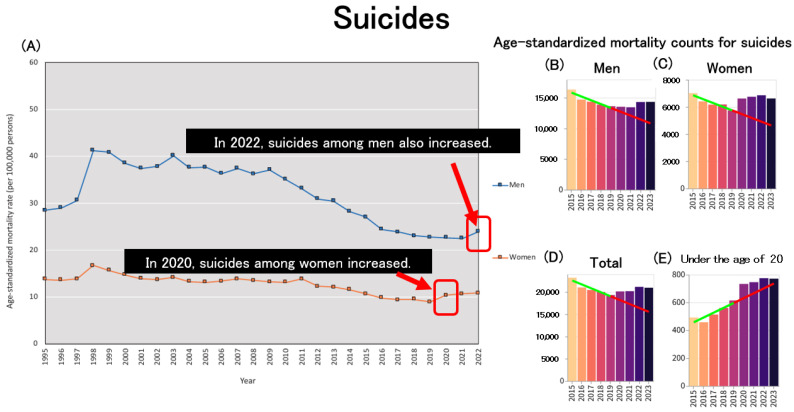
Trends in age-standardized mortality due to suicide. (**A**) Age-standardized mortality rates for suicide across Japan [2]. (**B**) Age-standardized mortality counts for suicide among men. (**C**) Age-standardized mortality counts for suicide among women. (**D**) Age-standardized mortality counts for suicide in both sexes. (**E**) Age-standardized mortality counts for suicide among individuals under 20 years of age, in both sexes. (**B**–**E**): The regression line was constructed based on data from the period up to 2019 [33].

**Table 1 healthcare-13-01305-t001:** URLs of the social media reporting the increase in the number of deaths in Japan (accessed on 27 May 2025).

**Increase in the number of deaths nationwide in Japan**
https://x.com/cr_cidp/status/1895037084777488589
https://x.com/sunsun_38/status/1895017200710496449
https://x.com/donkey1399/status/1895646280887808434
https://x.com/MNHR_Labo/status/1921013263002705977
**Increase in the number of deaths nationwide and in each municipality in Japan in January 2025**
https://x.com/VPIbflbSdnuQKaw/status/1894371517703893129
https://x.com/donkey1399/status/1905063278776344895
https://x.com/MNHR_Labo/status/1905465631832740159
https://x.com/JINKOUZOUKA_jp/status/1904779750935323086
https://x.com/JINKOUZOUKA_jp/status/1896793205867491626
https://x.com/JINKOUZOUKA_jp/status/1896485581271838850
https://x.com/JINKOUZOUKA_jp/status/1892790468359966987

**Table 2 healthcare-13-01305-t002:** Area, population, population density, aging rate, and URL of population dynamics statistics site for each municipality (as of 2025).

City/Area	Area (km^2^)	Estimated Population (as of 1 February 2025)	Population Density (People/km^2^)	Aging Rate (%)	URLs of Population Dynamics Statistics Site for Each Municipality (accessed on 27 May 2025)
**Entire Japan**	377,974	123,779,000	334	29.3	https://www.mhlw.go.jp/toukei/saikin/hw/jinkou/kakutei23/dl/04_h2-1.pdf
					https://www.mhlw.go.jp/toukei/saikin/hw/jinkou/geppo/s2024/dl/202412.pdf
					https://www.stat.go.jp/data/jinsui/new.html
					https://www.stat.go.jp/data/jinsui/2023np/index.html
**Tokyo** **23 Wards**	628	9,876,493	15,739	21.0	https://www.hokeniryo.metro.tokyo.lg.jp/kiban/chosa_tokei/jinkodotaitokei/kushityosonbetsu
					https://www.toukei.metro.tokyo.lg.jp/juukiy/2025/jy25000001.htm
**Yokohama**	438	3,766,732	8595	25.0	https://www.city.yokohama.lg.jp/city-info/yokohamashi/tokei-chosa/portal/jinko/dotai/nen/
					https://www.city.yokohama.lg.jp/city-info/yokohamashi/tokei-chosa/portal/jinko/dotai/tsuki.html
					https://www.city.yokohama.lg.jp/city-info/yokohamashi/tokei-chosa/portal/jinko/choki.html
**Osaka**	225	2,794,005	12,399	25.8	https://www.city.osaka.lg.jp/kenko/
					https://www.pref.osaka.lg.jp/documents/4370/r04c05.pdf
					https://www.city.osaka.lg.jp/toshikeikaku/page/0000541634.html
					https://www.city.osaka.lg.jp/toshikeikaku/
**Nagoya**	326	2,331,413	7141	25.0	https://www.city.nagoya.jp/somu/page/0000051424.html
					https://www.city.nagoya.jp/kenkofukushi/page/0000176348.html
					https://www.city.nagoya.jp/somu/page/0000181516.html

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
