# Peer review of "Unraveling Rising Mortality: Statistical Insights from Japan and International Comparisons"

_healthcare, 2025, doi:10.3390/healthcare13111305_

Round 1
Reviewer 1 Report
Comments and Suggestions for Authors
Commentary Review
Unraveling Rising Mortality: Statistical Insights from Japan and International Comparisons.
Reviewer’s Comment:
I appreciate the authors' efforts in addressing a highly relevant topic—namely, excess mortality in Japan in the context of the COVID-19 pandemic. The comparative analysis with other countries adds significant value to the manuscript. However, I believe it is necessary to delve deeper into the possible underlying causes of the observed increase in mortality beginning in 2021.
Overall, the manuscript is excellent in both content and clarity of presentation. Nevertheless, I would like to suggest that the authors offer a more detailed discussion of certain clinical and epidemiological aspects that may be associated with the outcomes described.
In particular, it would be valuable to explore whether a subset of the population that survived COVID-19—especially individuals with cardiovascular risk factors, advanced age, sarcopenia, and frailty—may have developed subsequent complications that contributed to the increased mortality. Additionally, it would be pertinent to consider the potential impact of corticosteroid use during the acute phase of illness, and how this treatment may have altered metabolic and immune responses, possibly facilitating the development of conditions such as post-COVID diabetes mellitus. This, in turn, could increase the risk of cardiovascular events and medium-term mortality.
Furthermore, I suggest that the authors:
Expand the discussion on the proposed multifactorial contributors, particularly those related to access to healthcare and the indirect effects of the pandemic—such as social, economic, and psychological stress. Including quantitative or qualitative data to support these claims, if available, would strengthen the argument.
Provide a more detailed analysis of the impact of non-communicable diseases, particularly cardiovascular disease and cancer, and how their diagnosis and treatment were affected during the post-pandemic period. It would be helpful to know whether data exist on delayed diagnoses, treatment interruptions, or other relevant clinical indicators.
I believe that incorporating these elements would greatly enrich the manuscript and support a more robust discussion that could inform the development of public health policies aimed at reducing overall mortality and preventable cardiovascular events in vulnerable populations, such as those described by the authors.
Author Response
Reviewer 1
I appreciate the authors' efforts in addressing a highly relevant topic—namely, excess mortality in Japan in the context of the COVID-19 pandemic. The comparative analysis with other countries adds significant value to the manuscript. However, I believe it is necessary to delve deeper into the possible underlying causes of the observed increase in mortality beginning in 2021.
Overall, the manuscript is excellent in both content and clarity of presentation. Nevertheless, I would like to suggest that the authors offer a more detailed discussion of certain clinical and epidemiological aspects that may be associated with the outcomes described. 
- I sincerely appreciate your favorable evaluation of my manuscript and am truly grateful for your precise and valuable suggestions.
In particular, it would be valuable to explore whether a subset of the population that survived COVID-19—especially individuals with cardiovascular risk factors, advanced age, sarcopenia, and frailty—may have developed subsequent complications that contributed to the increased mortality. Additionally, it would be pertinent to consider the potential impact of corticosteroid use during the acute phase of illness, and how this treatment may have altered metabolic and immune responses, possibly facilitating the development of conditions such as post-COVID diabetes mellitus. This, in turn, could increase the risk of cardiovascular events and medium-term mortality.
- I have addressed the points you raised regarding older age, sarcopenia, frailty, and the use of corticosteroids in treatment in Lines 390–407. I would appreciate it if you could kindly review this section.
Furthermore, I suggest that the authors:
Expand the discussion on the proposed multifactorial contributors, particularly those related to access to healthcare and the indirect effects of the pandemic—such as social, economic, and psychological stress. Including quantitative or qualitative data to support these claims, if available, would strengthen the argument.
- Regarding the point you raised about the indirect effects—such as healthcare access and socio-psychological issues—we have addressed this by discussing discrimination against healthcare workers and restrictions on hospital visits. Please refer to Lines 421–437 for these additions.
Provide a more detailed analysis of the impact of non-communicable diseases, particularly cardiovascular disease and cancer, and how their diagnosis and treatment were affected during the post-pandemic period. It would be helpful to know whether data exist on delayed diagnoses, treatment interruptions, or other relevant clinical indicators.
- Regarding cardiovascular diseases, we have provided a description in Lines 353–369. Concerning delays in cancer diagnosis, this has been addressed in Lines 370–381. I would appreciate it if you could kindly review these sections.
I believe that incorporating these elements would greatly enrich the manuscript and support a more robust discussion that could inform the development of public health policies aimed at reducing overall mortality and preventable cardiovascular events in vulnerable populations, such as those described by the authors.
- Thank you very much for your insightful and valuable comments. I sincerely appreciate your continued support.

Reviewer 2 Report
Comments and Suggestions for Authors
This article evaluates the increase in mortality rates in Japan both domestically and internationally. It addresses the increased mortality after the COVID-19 pandemic in a multidimensional manner: various factors such as aging, problems accessing the healthcare system, and possible effects of vaccination are taken into account. The language of the article is clear and understandable. However, there are significant deficiencies in terms of methodology and use of resources.
1) The article type should be written as an epidemiological research type. The article is understood as secondary data analysis. It is presented as a compilation.
2) Why were social media sources needed? It weakens the power of the article. It is recommended to remove it.
3) There is no Statistical Analysis section in the article. Statistical analysis should be done and added. Rates should be given standardized according to age and gender, and the standardization method should be specified.
4) Mortality rates should be estimated with regression analyses, and the model should be mathematically verified.
5) Factors affecting the increase in mortality should be evaluated with multivariate analysis methods: age, vaccination status, access to the health system, social security system, life expectancy, etc.
6) Figure 2 shows the number of deaths. These figures should be standardized rates. Even if population growth is constant, raw figures can be misleading.
7) Figure 3 should be given as a standardized death rate, not a crude death rate.
The article addresses an important issue. However, the shortcomings in the methodology are acceptable provided that social media sources are reduced or removed and statistical analyses are strengthened.
Author Response
Reviewer 2
This article evaluates the increase in mortality rates in Japan both domestically and internationally. It addresses the increased mortality after the COVID-19 pandemic in a multidimensional manner: various factors such as aging, problems accessing the healthcare system, and possible effects of vaccination are taken into account. The language of the article is clear and understandable. However, there are significant deficiencies in terms of methodology and use of resources.
- The article type should be written as an epidemiological research type. The article is understood as secondary data analysis. It is presented as a compilation.
This study is not based on original data collected by the authors through primary epidemiological research. Rather, as you point out, it is a secondary analysis that synthesizes statistical data published by public institutions and local governments. In addition, relevant peer-reviewed literature and publicly available documents from official agencies, as well as reliable domestic and international statistical sources, have been cited. Given that the manuscript includes a substantial component of the authors' interpretations and perspectives, it is presented in the format of a review article rather than an original research article. I kindly ask for your understanding regarding these aspects.
2) Why were social media sources needed? It weakens the power of the article. It is recommended to remove it.
The author had anticipated that such opinions would arise prior to submission of this manuscript. The inspiration for this study originated from growing discussions on social media regarding the observed increase in all-cause mortality across Japan following the COVID-19 pandemic. Initially, the author did not consider these social media claims to be credible. However, upon creating graphs based on statistical data published by official institutions and local governments (see Fig. 1A), a trend line resembling those circulating on social media emerged, which the author found unexpectedly striking.
A considerable number of social media users have expressed suspicion regarding a potential association between increased mortality and COVID-19 vaccination. Although, this phenomenon is likely not unique to Japan, in the Japanese context, it has been difficult to openly question or critique public health measures—including vaccination—promoted by infectious disease experts and the government, due to the social stigma associated with dissenting views.
As discussed in this paper, the post-pandemic increase in mortality cannot be sufficiently explained by population aging or COVID-19 infection alone. Nonetheless, there remains a notable scarcity of peer-reviewed academic publications addressing alternative factors—such as the possible impact of vaccination—using formal scientific approaches. This gap in the literature is one of the reasons the author has referred to social media sources in Table 1. It should be noted, however, that while social media accounts are mentioned for context, no figures or graphs from social media were directly reproduced in this article. All figures, including those resembling those found online, were independently generated by the author based on publicly available raw data.
In light of the above points, I have included a discussion on the significance of presenting social media as an information source in Lines 566–599. I would appreciate it if you could kindly review this section and understand our rationale for including social media sources in the present manuscript.
- There is no Statistical Analysis section in the article. Statistical analysis should be done and added. Rates should be given standardized according to age and gender, and the standardization method should be specified.
- As rightly pointed out, mortality rates should ideally be age- and sex-standardized. However, the author does not have access to the detailed data necessary to perform such standardization. Furthermore, this manuscript is not presented as an original research article but rather takes the form of a narrative review. Consequently, statistical analyses were not conducted, and no dedicated section for statistical methods has been included. I ask for your understanding in this regard.
Specifically, I was unable to access mortality data stratified by age groups, which is essential for calculating age- standardized mortality rates. Therefore, I could not compute such rates myself. Instead, I have cited a peer-reviewed article that presents nationwide all-cause age-standardized mortality trends in Japan (J Epidemiol. 2025;35(3):154–159), and this reference is now included as Figure 4.
In addition, we have added a description of all-cause age-standardized mortality rates (ASMRs) in Lines 229–247, and included a new Figure 4. We kindly ask you to review these additions.
4) Mortality rates should be estimated with regression analyses, and the model should be mathematically verified.
As you pointed out, I created a regression model based on the crude mortality rates from the pre-pandemic period (2015–2019), as shown in Figure 3A. Figure 3B presents the differences between the predicted and observed crude mortality rates over time. A description of the regression model has been added in Lines 186–193. We kindly ask you to review this section.
5) Factors affecting the increase in mortality should be evaluated with multivariate analysis methods: age, vaccination status, access to the health system, social security system, life expectancy, etc.
Regarding the point you raised, I have addressed it as a limitation in the final conclusion section. Please kindly refer to Lines 707–713 and Lines 719–722 for the relevant descriptions.
6) Figure 2 shows the number of deaths. These figures should be standardized rates. Even if population growth is constant, raw figures can be misleading.
Regarding the time trends of the number of deaths in each municipality shown in Figure 2, we were unable to access age-specific mortality data for each municipality and therefore could not calculate age-standardized mortality rates. Consequently, similar to Figure 3, we decided to present the temporal changes in the differences between the observed values and those estimated by the regression model. Please refer to Lines 166–175 for the relevant description, and kindly review panels (B) Tokyo 23 Wards, (D) Yokohama, (F) Osaka, and (H) Nagoya in Figure 2.
7) Figure 3 should be given as a standardized death rate, not a crude death rate.
Regarding Figure 3, please refer to Lines 186–193 for the description of the regression model, as mentioned above.
The article addresses an important issue. However, the shortcomings in the methodology are acceptable provided that social media sources are reduced or removed and statistical analyses are strengthened.
I kindly ask for your understanding regarding the issues related to social media sources. Regarding the reliability of the independent website used in this article (https://medicalfacts.info/mortality.rb), which draws on official statistics published by public institutions, I have cross-checked its data against those published in J Epidemiol. 2025; 5:154–159. For example, comparisons can be seen between Figure 1A and Figure 4B, Figure 9A and 9B, as well as Figure 10A and Figures 10B and 10C. I sincerely appreciate your very valuable comments and thank you once again.

Reviewer 3 Report
Comments and Suggestions for Authors
The authors analyze the recent rise in mortality rates in Japan in comparison to data from Western countries and the United States and South Korea, investigating patterns and the correlation between COVID-19 pandemic statistics and vaccination rates. Nevertheless, given this paper concentrates on the recent escalation of mortality rates in Japan, it is imperative to examine trends in mortality rates that are not associated with COVID-19. Also, while the authors make their points in this paper, including objective data at the same time could make their claims more convincing, showing that the paper needs more editing.
Major comments
Incorporating data on the aging rates of each city in Table 2 would augment readers' comprehension of Figure 3.
Concerning Osaka, could the authors furnish evidence indicating that income inequalities are more pronounced than in the other three cities, that health risks and disparities in access to healthcare are escalating among low-income populations, and that the incidence of lifestyle-related diseases such as diabetes, obesity, and smoking is elevated?
Concerning the discourse in Figure 5, please elucidate the discussion pertaining to “Factor X” with greater specificity.
The authors indicate that the policy of minimizing interpersonal interaction to mitigate COVID-19 transmission, which resulted in a decline in routine health examinations, may have contributed to a rise in overall death rates. Is this a direct causal relationship, however? Please furnish explicit information regarding the tendencies of senior individuals to eschew medical care following the COVID-19 epidemic.
The authors claim that the Novavax protein-based COVID-19 vaccine (NVX-CoV2373) has fewer side effects than mRNA vaccines and suggest that the higher use of mRNA vaccines in Japan compared to Europe and America might be linked to higher death rates. Please furnish concrete proof demonstrating that NVX-CoV2373 exhibits a reduced incidence of adverse effects that may result in mortality when compared to mRNA vaccines.
Please elucidate the evidence substantiating the authors' assertion that COVID-19 countermeasures may have exacerbated the susceptibility of the elderly and diminished their immune function.
The authors assert that the extended enforcement of strict infection control measures in Japan has had economic and social repercussions, resulting in a rise in suicides within the country. Please provide evidence that supports this assertion.
The authors identify factors associated with the COVID-19 pandemic and vaccination as the primary contributors to the recent rise in mortality in Japan. What is the trend in mortality rates from non-infectious disease causes? It is essential to analyze the trends in national mortality causes to address the rise in mortality rates.
Minor comments
In Table 1, please differentiate text from URLs by employing bold font or alternative methods.
Reference 57 suggests that the annual count of 30,000 COVID-19 fatalities reported recently may be an underrepresentation due to a shift in Japan's reporting methodology from comprehensive reporting to an alternative system.
Author Response
Reviewer 3
The authors analyze the recent rise in mortality rates in Japan in comparison to data from Western countries and the United States and South Korea, investigating patterns and the correlation between COVID-19 pandemic statistics and vaccination rates. Nevertheless, given this paper concentrates on the recent escalation of mortality rates in Japan, it is imperative to examine trends in mortality rates that are not associated with COVID-19. Also, while the authors make their points in this paper, including objective data at the same time could make their claims more convincing, showing that the paper needs more editing.
Major comments
Incorporating data on the aging rates of each city in Table 2 would augment readers' comprehension of Figure 3.
Please refer to Table 2 for the aging rate data of each city (highlighted in red).
Concerning Osaka, could the authors furnish evidence indicating that income inequalities are more pronounced than in the other three cities, that health risks and disparities in access to healthcare are escalating among low-income populations, and that the incidence of lifestyle-related diseases such as diabetes, obesity, and smoking is elevated?
A description of the characteristics of Osaka has been added in Lines 206–222. Please kindly review this section.
Concerning the discourse in Figure 5, please elucidate the discussion pertaining to “Factor X” with greater specificity.
Details regarding Factor X have been specifically described in Lines 273–284. Please kindly review this section.
The authors indicate that the policy of minimizing interpersonal interaction to mitigate COVID-19 transmission, which resulted in a decline in routine health examinations, may have contributed to a rise in overall death rates. Is this a direct causal relationship, however? Please furnish explicit information regarding the tendencies of senior individuals to eschew medical care following the COVID-19 epidemic.
We consider that there is a direct causal relationship regarding this issue. A description of this matter has been added in Lines 338–350. Please kindly review this section.
The authors claim that the Novavax protein-based COVID-19 vaccine (NVX-CoV2373) has fewer side effects than mRNA vaccines and suggest that the higher use of mRNA vaccines in Japan compared to Europe and America might be linked to higher death rates. Please furnish concrete proof demonstrating that NVX-CoV2373 exhibits a reduced incidence of adverse effects that may result in mortality when compared to mRNA vaccines.
The fact that NVX-CoV2373 has fewer adverse reactions than mRNA vaccines is mentioned in Line 116, with reference 26 cited. Reference 26 is also cited in Line 620. Please kindly confirm these points.
Please elucidate the evidence substantiating the authors' assertion that COVID-19 countermeasures may have exacerbated the susceptibility of the elderly and diminished their immune function.
The possibility that worsening frailty in the elderly led to a decline in immune function is described in Lines 385–388. Please kindly review this section.
The authors assert that the extended enforcement of strict infection control measures in Japan has had economic and social repercussions, resulting in a rise in suicides within the country. Please provide evidence that supports this assertion.
The issue of suicide is addressed in Lines 643–663, and a new Figure 10 has been added. Please kindly review these additions.
The authors identify factors associated with the COVID-19 pandemic and vaccination as the primary contributors to the recent rise in mortality in Japan. What is the trend in mortality rates from non-infectious disease causes? It is essential to analyze the trends in national mortality causes to address the rise in mortality rates.
- Matters related to non-communicable diseases are discussed in Lines 445–488, with new Figures 7 (cardiovascular diseases), 8 (malignant neoplasms), and 9 (senility) included. Please kindly review these sections.
Minor comments
In Table 1, please differentiate text from URLs by employing bold font or alternative methods.
- We have bolded the text to enhance its visibility. Please kindly review.
Reference 57 suggests that the annual count of 30,000 COVID-19 fatalities reported recently may be an underrepresentation due to a shift in Japan's reporting methodology from comprehensive reporting to an alternative system.
Regarding this point, we have discussed it in Lines 497–505. Please kindly review. We sincerely appreciate your very valuable comments and thank you once again.

Round 2
Reviewer 2 Report
Comments and Suggestions for Authors
The explanations given by the author can be considered sufficient.
Reviewer 3 Report
Comments and Suggestions for Authors
The authors adequately addressed the reviewer's comments. Consequently, the paper's worth to readers has been augmented. The reviewer endorses the publishing of the revised manuscript.